## RESEARCH ARTICLE

# Elucidating the genetic architecture of migratory timing in a songbird migrant, the great reed warbler, *Acrocephalus arundinaceus*

Emily R. Fackler[1,*], Dmitry Kishkinev[2], Petr Procházka[3] and Robert R. Fitak[1]

## ABSTRACT

Great reed warblers (*Acrocephalus arundinaceus*) have become an important species for understanding long-distance avian migration, yet the genetic basis of their migratory timing remains unknown. While previous studies have identified candidate genes influencing migration timing in other species, their role in great reed warblers remains unexplored. Additionally, it is unclear whether the genetic basis of migratory timing differs between spring and autumn migrations. This study aims to uncover genetic factors influencing migration timing, providing insights into the evolutionary and ecological processes shaping long-distance migration. We conducted pooled whole-genome sequencing representing four great reed warbler migratory chronotypes: early spring, late spring, early autumn, and late autumn. By comparing $F_{ST}$ and allele frequency differences, we determined that the spring migration had a larger genetic contribution than the autumn migration; however, the effect sizes were small (0.03 and 0.001, respectively). When comparing the early and late spring pools, we identified 93 candidate genes enriched for functions related to lipid hydrolysis that putatively influence great reed warbler migratory behavior. Our results provide insight into the genetic differentiation underlying migratory timing in great reed warblers, which is crucial for predicting how they will adapt to shifting environmental conditions due to climate change and habitat loss.

KEY WORDS: Bird migration, Genomics, Pool-seq, Chronotype, Lipid hydrolysis, Whole-genome sequencing

## INTRODUCTION

Numerous species of birds are well-known for their long, seasonal migrations in both the spring and autumn, however, there is a lack of knowledge on the mechanisms that control the timing of these migrations. The innate response to migrate at the same time each year, as a part of the circannual rhythm, is controlled by an endogenous circannual clock in all migratory organisms (Gwinner,

1996). Environmental cues such as photoperiod and light intensity determine circannual rhythms in birds (Bentley et al., 1998; Gwinner, 2003; Åkesson et al., 2017). Birds perceive daily changes in these cues as signals to determine the season and time of day thus informing them when to begin their migratory journeys. While these cues may signal that it is time to migrate, individual birds show variation in their departure timing. This variation can be categorized into distinct migratory chronotypes, with some individuals migrating earlier, later, or at an intermediate time (Åkesson and Helm, 2020; Bossu et al., 2022).

Bird migration is a complex phenomenon that is most likely influenced by both internal and external factors. Previous studies in birds have demonstrated that the variation in both spring and autumn migratory chronotypes may result from a variety of environmental elements and cues (Visser et al., 2010; Åkesson et al., 2017). For example, differences in the amount of rainfall from year to year has been found to influence when American redstarts (*Setophaga ruticilla*) start their spring migration (Studds and Marra, 2011). Along similar lines, Cooper et al. (2015) found that food-reduced American redstarts had a later departure date for the spring migration when compared to a control group. The amount of vegetation in wintering sites have been shown to influence when barn swallows (*Hirundo rustica*) arrive at their breeding site and, therefore, when they depart from their wintering site (Saino et al., 2004). Mitchell et al. (2012) found that autumn migratory departure dates were due to weather conditions and the completion of breeding in adult Savannah sparrows (*Passerculus sandwichensis*). Although these factors influence when a bird is ready to start its migratory journey, many bird species still exhibit migratory restlessness and seasonal migratory disposition even when deprived of external environmental cues (Berthold, 1984). This suggests that the innate response to migrate is not just dictated by external environmental factors but also governed by endogenous biological clocks, which could be genetically controlled (Liedvogel et al., 2011).

Great reed warblers (*Acrocephalus arundinaceus*) are common, dietary generalists and habitat specialist passerine birds and one of the largest members of the family Acrocephalidae. The species is widespread among breeding sites throughout Eurasia and migrates to wintering areas in sub-Saharan Africa (Bensch et al., 1998). Great reed warblers mostly migrate between dusk and dawn and have been known to reach altitudes as high as 6458 m a.s.l. (Liechti et al., 2018; Sjöberg et al., 2018; Sjöberg et al., 2021). They have become one of the best-known long-distance migrants in the African-Palearctic migration system, making it an important species for studying migratory behavior and timing (Lemke et al., 2013; Horns et al., 2016; Koleček et al., 2016; Hasselquist et al., 2017; Brlík et al., 2020; Emmenegger et al., 2021; Malmiga et al., 2021; Sjöberg et al., 2021). The average departure dates, arrival dates, duration, and speed for both pre-breeding (i.e. spring) and post-breeding (i.e.

[1]Department of Biology, Genomics and Bioinformatics Cluster, University of Central Florida, Orlando, FL 32816, USA. [2]Faculty of Natural Sciences, School of Life Sciences, Keele University, Newcastle-under-Lyme, Staffordshire ST5 5BG, UK. [3]Institute of Vertebrate Biology of the Czech Academy of Sciences, Květná 8, 603 65 Brno, Czech Republic.

*Author for correspondence (emily.fackler@ucf.edu)

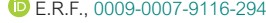 E.R.F., 0009-0007-9116-2943

autumn) migrations across different great reed warbler populations are highly consistent (Koleček et al., 2016). Despite the consistency among populations, the departure and arrival dates of individuals within a population are highly variable (Lemke et al., 2013; Stępniewska et al., 2020); suggesting that great reed warblers, like other migratory birds, exhibit differences in their migratory chronotypes.

Previous research has suggested that there may be a selective advantage to start the spring migration early because arriving early at the breeding site often results in higher reproductive success (Dyrcz, 1986; Ezaki, 1990; Bensch and Hasselquist, 1991; Hasselquist, 1998; Nilsson et al., 2013; Tarka et al., 2015). In autumn there is arguably less reproductive advantage to reaching the wintering site early, although variation in timing within the autumn migration could still be a result of complex environmental factors such as weather and food availability/habitat suitability at stopovers (Nilsson et al., 2013). Additionally, due to carryover effects, autumn migratory chronotype could be a result of when breeding is completed (Nilsson et al., 2013; Chmura et al., 2020; Imlay et al., 2021). In great reed warblers (and other avian species), the difference between spring and autumn migration speed is consistent with this theory, as spring migration has been documented as 62% faster (220 km/day) than in autumn (139 km/day) and with less than half the number of days at stopover sites (13.3 and 27.1 days, respectively) (Lemke et al., 2013). Autumn migration in great reed warblers is also particularly complex since great reed warblers shift migration strategies en route (Stępniewska et al., 2020). They start their autumn migration with multiple, short stops with low energy reserves and end with one long, well-fueled flight (Stępniewska et al., 2020). These differences in migration speed and strategy suggest that timing is under stronger selection pressure in spring, likely due to its impact on reproductive success (Hasselquist, 1998; Kokko, 1999). This, in turn, implies that genetic factors play a larger role in determining spring migratory timing compared to autumn migration, where external environmental conditions may have a greater influence.

Migratory behavior is a complex trait and likely influenced by many genes of varying effect. Nevertheless, multiple genes have been regularly linked with migratory behaviors in birds. The best-studied examples include length polymorphisms in the *Adcyap1* and *Clock* genes which are associated with increased migratory restlessness (i.e. 'Zugunruhe') and circannual timing, respectively (Mueller et al., 2011; Ralston et al., 2019; Bingman and Ewry, 2020; Justen et al., 2022; Le Clercq et al., 2023; Sharma et al., 2023). However, in a variety of trans-Saharan migratory birds, including great reed warblers, *Clock* gene diversity was associated with migratory timing, whereas size polymorphisms in both *Clock* and *Adcyap1* lacked any significant associations (Bazzi et al., 2016). This suggests that there are likely other genes associated with migratory chronotypes beyond these and a handful of other candidate genes. Bossu et al. (2022) explored this question in American kestrels (*Falco sparverius*) by using a genomics approach and found that three genes related to migration and circadian rhythms, *Top1*, *Peak1*, and *Cpne4*, had strong associations with migratory chronotypes. Nevertheless, little is known whether these same genes or other genes influence the migratory chronotypes of great reed warblers.

Due to the lack of knowledge on whether migratory chronotype has a genetic basis and the extent of its influence in great reed warblers, we investigated the genetic contribution to migratory chronotype in a Czech population of great reed warblers using a whole-genome approach. In this study, we (1) compared the level of genetic contribution between the spring and autumn migrations and (2) identified candidate genes associated with the migratory timing of great reed warblers by comparing genetic differentiation between early and late spring chronotypes. We hypothesized that (1) the spring migration would have a larger genetic contribution than the autumn migration because the spring migration offers a larger reproductive advantage, (2) polymorphisms in circadian related genes such as *Clock*, *Adcyap1*, *Top1*, *Peak1*, and *Cpne4* would be associated with the migratory chronotype, and (3) there are additional candidate genes that influence the migratory timing of great reed warblers. Our results provide insight into how genes influence migratory timing behavior in great reed warblers and give us a better understanding of avian migration and phenology.

## RESULTS

Four pools, each with nine great reed warblers, were generated: early spring, late spring, early autumn, and late autumn. The birds within and across each pool represented a matched set of individuals at the outer fringes of the range of seasonal migratory timing (Fig. 1). On average, the males in the early spring pool departed 25 days before the males in the late spring pool and the females in the early spring pool departed 20 days before the females in the late spring pool. Similarly, the males in the early autumn pool departed 18 days before the males in the late autumn pool, while the females in the early autumn pool departed 25 days before the females in the late autumn pool. The sex ratio between the early spring, late spring, and early autumn pools were the same (three females: six males) while the late autumn pool was slightly different (four females: five males) in order to account for the number of nests, departure year, and whether the individual successfully bred.

Each pool was sequenced to a mean depth of coverage of 31.8X, and after quality filtering and mapping to the reference genome the mean depth of coverage was 22.4X. A summary of the sequencing reads and bases before filtering (raw), after filtering, and mapped to the reference genome are provided in Fig. S1. Using the pool-seq software *PoPoolation2*, a total of 14,439,829 polymorphic sites were identified across the genome. The $F_{ST}$ calculated at these polymorphic sites between the early spring versus late spring pairwise comparison was used to define 1,156,611 windows across the great reed warbler with a genome-wide mean $F_{ST}$ of 0.052 (s.d.=0.034).

### Quantifying the seasonal genetic contribution

Across more than 14 million polymorphic sites, the $F_{ST}$ between the early spring versus late spring comparison ($M$=0.050, s.d.=0.058) was significantly larger than in the early autumn versus late autumn comparison ($M$=0.047, s.d.=0.055) ($Z$=118.8, $P$<2.2×10$^{-16}$). A similar pattern was also found in allele frequency differences, where the difference between the early spring versus late spring comparison ($M$=0.110, s.d.=0.107) was significantly larger than in the early autumn versus late autumn comparison ($M$=0.109, s.d.=0.102) ($Z$=25.8, $P$<2.2×10$^{-16}$). Although significant, the mean differences in both FST (95% CI 0.0024–0.0025) and allele frequency differentiation (0.00093–0.0011) are very small. This suggests that the genetic contribution to spring versus autumn migration timing is unlikely to be biologically meaningful.

### Candidate genes associated with spring migratory chronotype

Using $F_{ST}$ spline-based defined windows, a total of 229 windows contained at least two significant SNPs (Fisher's exact test FDR <0.01) (Fig. 2). The mean $F_{ST}$ of these outlier windows ($M$=0.22, s.d.=0.096) was 4.2 times higher than the genome-wide

Biology Open

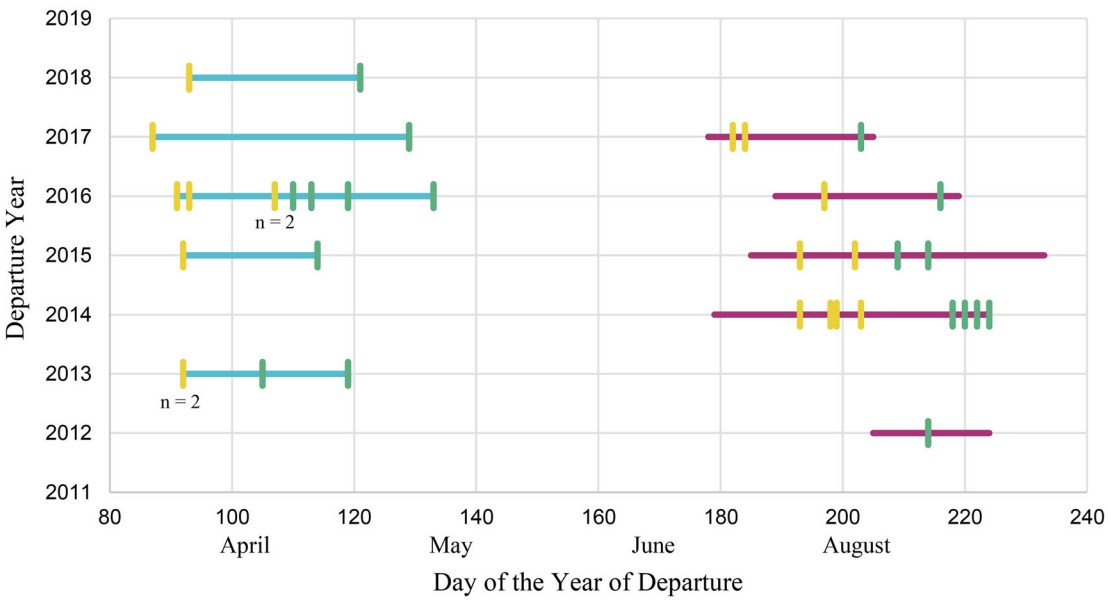

**Fig. 1. The day of the year that each warbler started its spring and autumn migration.** The x-axis is the day of the year that each individual departed (started its migration) from either the breeding or wintering site. The y-axis is the year that the geolocator data was gathered. The blue horizontal lines show the range of the spring migration departure dates across all 40 birds. The purple horizontal lines show the range of the autumn migration departure dates across all 54 birds. The vertical lines represent the warblers chosen for the early pools (yellow) and the late pools (green). The notation n=2 means that there were two individuals who had that departure date, so the vertical line represents two individuals.

mean $F_{ST}$ and represented a combined total of 479,959 bp. These outlier windows corresponded to 93 protein-coding genes that differed significantly between the early and late spring migrants (Table S2).

There were no significantly overrepresented GO terms in the PANTHER Pathways, PANTHER GO-Slim Biological Process, PANTHER GO-Slim Cellular Component, and PANTHER Protein Class categories for the 93 protein-coding genes. However, there were two significantly overrepresented GO terms for the PANTHER GO-Slim Molecular Function category. The GO term triglyceride lipase activity had a 45.1-fold enrichment (Fisher's exact test, FDR=8.3×10$^{-4}$) and the GO term carboxylic ester hydrolase activity had a 10.8-fold enrichment (Fisher's exact test, FDR=2.9×10$^{-4}$) (Fig. 3). Four of the 93 genes were associated with the triglyceride lipase activity GO term (*Pnlip*, *Pnliprp1*, *Pnliprp2*,

and *Pnliprp3*) and five were associated with the carboxylic ester hydrolase activity GO term (*Pnlip*, *Pnliprp1*, *Pnliprp2*, *Pnliprp3*, and *Aoah*). Some additional PANTHER Molecular Function GO terms assigned to the 93 candidate genes were RNA polymerase II cis-regulatory region sequence-specific DNA binding (*Egr2*, *Dmrt1*, *Mynn*, and *Mafa*), RNA binding (*U2surp* and *Rbm18*), protein kinase binding (*Dok7* and *Rb1cc1*), and protein serine/threonine kinase activity (*Rps6ka2* and *Prkcq*). The common PANTHER Biological Process GO terms were canonical Wnt signaling pathway (*Fzd4*), negative regulation of Wnt signaling pathway (*Dact1* and *Nxn*), and regulation of transcription by RNA polymerase II (*Pcgf5*, *Egr2*, *Mynn*, *Ctbp2*, and *Mafa*). PANTHER Protein Class GO terms commonly assigned were cysteine protease (*Capn5*, *Atg4a*, and *Zranb1*), guanyl-nucleotide exchange factor (*Mcf2l_0*, *Arhgef40*, and *Arhgef16*), RNA methyltransferase

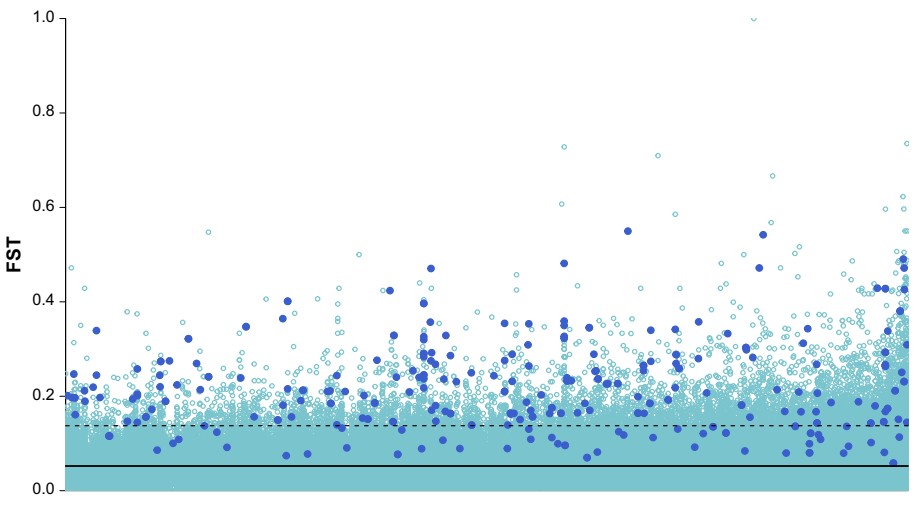

**Fig. 2. Mean $F_{ST}$ values for each window in the early spring versus late spring comparison.** The light blue spots are windows with less than two significant SNPs (Fisher's exact test; downsampled to 25% for visual purposes) and the dark blue spots are windows with at least two significant SNPs (i.e. outlier windows). The genome-wide mean $F_{ST}$ across all windows (solid black line) and the upper 97th percentile (dashed black line) are shown. The scaffolds are arranged in numerical order from VZST01000001.1 to VZST01029468.1. Due to the large number of scaffolds in the reference genome (29,468), they are not shown individually.

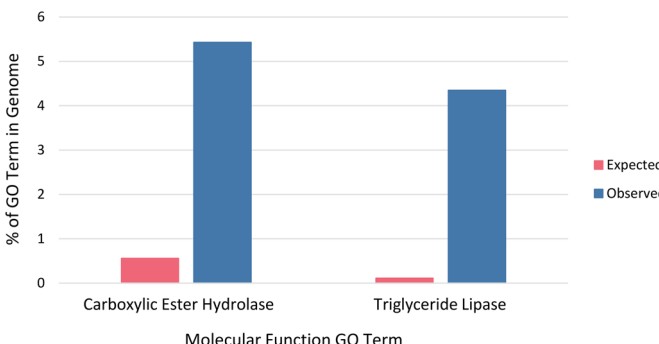

**Fig. 3. Overrepresented GO terms from PANTHER.** The pink columns are the percentage of genes with this GO term across the genome. The blue columns are the percentage of genes observed with these GO terms in the 93-candidate gene set.

(*Trmt5*, *Trmt1l*, and *Mettl25*), and scaffold/adaptor protein (*Prkn*, *Grb14*, *Rb1cc1*, and *Sh2b3*). Common PANTHER Pathway GO terms were angiogenesis (*Prkcq*, *Egf1*, *Grb14*, and *Pik3c2g*), Wnt signaling pathway (*Prkcq*, *Fzd4*, *Ctbp2_1*, and *Dact1*), EGF receptor signaling pathway (*Prkcq*, *Pik3c2g*, and *Nrg3*), and endothelin signaling pathway (*Prkcq*, *Pik3c2g*, and *Gnas*).

## DISCUSSION
### Genetic contribution to spring and autumn migration
We hypothesized that the spring migration would have a larger genetic contribution when compared to the autumn migration because there is a clear reproductive (i.e. fitness) advantage for great reed warblers to arrive at the breeding site early (Dyrcz, 1986; Ezaki, 1990; Bensch and Hasselquist, 1991; Hasselquist, 1998; Nilsson et al., 2013; Tarka et al., 2015). We found that both $F_{ST}$ and allele frequency differences were significantly higher in the spring comparison than in autumn. Although our results support this hypothesis, the effect size was very small (differences in mean $F_{ST}$ and allele frequencies were 0.003 and 0.001, respectively). This is consistent with migratory chronotype being a complex phenotype influenced by numerous loci of small effect rather than a few, large-effect loci (Bossu et al., 2022; de Greef et al., 2023). Additional studies that better quantify the genetic contribution across seasons using Bayesian sparse linear mixed models (de Greef et al., 2023) from paired whole-genome sequencing and high-throughput phenotyping are warranted for confirming or refuting these findings.

Because we aimed to select the most extreme migratory phenotypes while maintaining the sex ratio and ensuring that all the individuals in the autumn pools had only one nest and successfully bred, we were limited by the number of individuals we could pool. This resulted in a relatively limited, but comparable, sample size in each pool (*n*=9 individuals, i.e. 18 alleles per autosomal locus). It has been suggested to have a larger number of individuals in each pool (Schlötterer et al., 2014); however, Anand et al. (2016) showed that reliable allele frequency data can be obtained with a pool size of only 12 individuals. All pools in our study had a female:male sex ratio of 3:6 except for the late autumn pool which had a ratio of 4:5. This difference among pools is small, but notably the current great reed warbler reference genome available with annotation (accession ASM1339868v1) is a draft lacking assignment to specific chromosomes. It is possible that several of the outlier windows we identified may be on the Z or W sex chromosomes, which have different allele frequencies depending on the sex that was not accounted for in our analysis. Furthermore, a larger sample size would likely result in detecting

additional, but weaker differentiation. As a result, the loci we report here likely do not represent all the loci linked to migratory chronotype. Additional studies with larger samples and improved genomic resources for the great reed warbler and other migratory species are needed in the future to provide more robust results.

### Candidate genes associated with spring migratory chronotype
There were 93 candidate genes in or near regions that were significantly differentiated between the early and late spring chronotypes. The two GO terms that were overrepresented in this list of candidate genes were triglyceride lipase activity and carboxylic ester hydrolase activity. Triglyceride lipase activity refers to proteins that assist in catalyzing the reaction between triacylglycerol and water to produce diacylglycerol and carboxylate, and carboxylic ester hydrolase activity assists in the hydrolysis of a carboxylic ester bond (Tarka et al., 2015; Thomas et al., 2022). Both overrepresented GO terms are in the lipase gene family and participate in the breakdown or hydrolysis of lipids. More specifically, lipases hydrolyze lipids such as triglyceride by hydrolyzing ester bonds (Wong and Schotz, 2002; Dixit et al., 2022). Between the early and late spring migrants we saw significant differentiation in five genes involved in lipid hydrolysis (*Pnlip*, *Pnliprp1*, *Pnliprp2*, *Pnliprp3*, and *Aoah*). Notably, migratory birds go through a period of preparation before migration which involves the accumulation of lipids (Jenni-Eiermann and Jenni, 1996; Rani et al., 2017). These lipids are mainly in the form of high energy triglycerides that provide the bird with enough energy to complete the migration process (Araújo et al., 2019). These triglycerides are primarily stored in adipose tissue to support the bird and are broken down throughout its migratory journey (Dixit et al., 2022). Past studies have shown that every organ (with the exception of the heart), sees an increase in lipid content which makes the lipid accumulation 50–60% of a bird's body mass (Odum and Perkinson, 1951; Odum and Connell, 1956; Guglielmo, 2018). It is well known that stored fat content is directly associated with migratory behavior (Blem, 1990; Witter and Cuthill, 1993; Schmaljohann and Eikenaar, 2017). It has been shown that passerine migrants with larger subcutaneous fat stores exhibit higher levels of migratory restlessness (Farner, 1950; Fusani et al., 2009; Eikenaar and Schläfke, 2013; Sjöberg et al., 2015; Lupi et al., 2016). Further, great reed warblers have been shown to alter their autumn migration strategy depending on accumulated energy reserves (Stępniewska et al., 2020). Our results thus suggest that the early and late spring chronotypes of great reed warblers may have a different genetic predisposition in their ability to metabolize stored lipids when migrating. Bounas et al. (2023) identified 1167 differentially expressed genes when comparing garden warblers (*Sylvia borin*) in both a lean and fat state. Five of those differentially expressed genes (*Ankrd29*, *Nlgn4x*, *Btbd6*, *Rps6ka2*, and *Slc9a5*) overlap with the 93 candidate genes that differed significantly between the early and late great reed warbler spring migrants. Little is known about the relationship between these five genes and migration; however, He et al. (2015) found that *Ankrd29* was differentially expressed in grass carp (*Ctenopharyngodon idella*) that underwent compensatory growth. This further suggests that the ability to metabolize lipids and gain fat could play a part in determining great reed warbler migratory chronotype. Future work targeting the intersection of migratory chronotype, genetics, and functional ability to metabolize stored fats is needed to confirm this effect. It is possible that early and late chronotype individuals show differences in their lipogenesis rates.

For example, assuming equal dietary intake, early chronotype individuals may accumulate fat reserves faster due to a higher speed of fat deposition compared to late chronotype individuals. Additionally, it is possible that early and late chronotype individuals differ in their use of fat for thermoregulation, flight, feeding efforts, or combinations thereof.

Previous work in American kestrels identified candidate genes, including *Top1*, *Peak1*, and *Cpne4*, that were associated with migratory chronotype (Bossu et al., 2022). None of these genes overlapped the 93 candidate genes that differed between great reed warbler spring migratory chronotypes in our study. Genes such as *Top1* and *Cpne4* genes are considered clock-linked genes because they regulate the core clock genes. Past studies that investigated core clock genes in migratory birds have seen that these genes have varying levels of effect depending on the species (Dor et al., 2011; Mueller et al., 2011; Peterson et al., 2013; Krist et al., 2021). For example, Bazzi et al. (2015) found that variation in the *Clock* gene in barn swallows was associated with a delay in spring migration timing; whereas, research on painted buntings (*Passerina ciris*) found no correlation between migration timing and variation in the *Clock* and *Adcyap1* gene (Contina et al., 2018). This suggests that (1) there may be a taxon-specific relationship between clock-linked genes and migratory chronotypes lacking in great reed warblers indicating different underlying mechanisms, (2) the complete set of genes linked to circadian pathways and *Clock* are not yet fully understood, or (3) it could be a combination of both.

Beyond the possibility that migratory chronotypes are taxon-specific and are not associated with clock-linked genes in great reed warblers, there are multiple technical explanations for why we found no overlapping genes. First, Bossu et al. (2022) used a RAD-seq methodology that limited their analyses to only 1843 possible genes – a small fraction of the total genes in the genome and thus missing many potential candidate genes that may have matched our results. Second, the authors reported using a very relaxed set of detection criteria for defining outlier loci ($F_{ST} \geq$ the 90th percentile). This relaxed threshold was applied to detect what the authors predicted to be numerous loci of small effect. Lastly, for the initial detection of outlier loci, Bossu et al. (2022) calculated $F_{ST}$ between resident and migratory populations of American kestrel rather than migratory chronotypes (later, a targeted analysis of clock-linked genes was analyzed between early versus late chronotypes). Therefore, their initial set of candidate genes are more likely associated with propensity to migrate as opposed to migratory chronotype.

Since American kestrels belong to an entirely different order (Falconiformes) than great reed warblers (Passeriformes), intra-order, whole-genome comparisons are ideal. Indeed, a recent study in migratory purple martins (*Progne subis subis*), also of the order Passeriformes, examined genetic associations with migratory chronotype using whole-genome sequencing (de Greef et al., 2023). The authors compared the ten earliest and ten latest spring migrants and discovered a 2-Mb region on chromosome 1 associated with the purple martin's migratory chronotype. Interestingly, of the 13 genes in this single genomic region one gene, *Mettl25*, overlapped our dataset of 93 genes. The *Mettl25* gene codes for the protein methyltransferase-like 25 whose function is still unknown (Wong and Eirin-Lopez, 2021). However, the GO term 'enables methyltransferase activity' is assigned to *Mettl25* (https://www.genecards.org/; Stelzer et al., 2016). Phylogenetic examination of *Mettl25* has determined it is closely related to *Mettl25B*, which has dimethyltransferase activity (Wong and Eirin-Lopez, 2021; de Greef et al., 2023). In fact, the entire family of methyltransferase-like (*Mettl*) enzymes are known to catalyze the

transfer of methyl groups to DNA, RNA, and proteins (Wong and Eirin-Lopez, 2021). If the function of *Mettl25* is similar to *Mettl25B* due to their close homology, *Mettl25* could enable or activate methylation and therefore alter gene expression. Epigenetic processes, such as increased DNA methylation in the *Clock* gene, are already known to impact the timing of spring migration patterns (Saino et al., 2017; Merlin and Liedvogel, 2019). We also identified the gene *Dicer1* in our list of candidate genes. *Dicer1* is a highly conserved RNase enzyme that plays a pivotal role in processing small RNA molecules, such as microRNAs (miRNAs), in multiple RNA inference pathways (Foulkes et al., 2014). Differential expression of microRNAs was previously identified between migratory and non-migratory monarch butterflies (Zhan et al., 2011), and post-transcriptional regulation of gene expression via small RNAs and other mechanisms has been proposed as fundamental to future studies of migratory phenotypes (Merlin and Liedvogel, 2019). Collectively, our results further implicate epigenetic processes, including both methylation and small RNA interference, in influencing migratory chronotypes.

When comparing gene expression in garden warblers in both a lean and fat state, Bounas et al. (2023), found that genes related to angiogenesis were differentially expressed. One of those genes was *Epas1*, which has also been found to be differently expressed in other migratory species at different stages (Sharma et al., 2018; Frias-Soler et al., 2020). For example, *Epas1* was found to be differentially expressed in northern wheatears (*Oenanthe oenanthe*) during three different stages: lean, undergoing fattening, and at their maximal migratory body mass (Frias-Soler et al., 2020). Additionally, Sharma et al. (2018) found that *Epas1* was differentially expressed in the hypothalamus of black-headed buntings (*Emberiza melanocephala*) when comparing individuals in winter non-migratory and spring migratory states. These studies show that angiogenesis is potentially a part of the migratory preparation process for these avian species and great reed warblers do not appear to be an exception. Four of the 93 candidate genes (*Prkcq*, *Egf1*, *Grb14*, and *Pik3c2 g*) that differed significantly between the early and late great reed warbler spring migrants were assigned angiogenesis as their PANTHER Pathway. This suggests that the formation or growth of new blood vessels, amongst other things, could play a role in determining great reed warbler migratory chronotype. As Frias-Soler et al. (2022) suggests, these new blood vessels could allow for more oxygen to be carried throughout the bird's body, which would be beneficial during the migration process.

## Conclusion

In this study, we discovered that in the Czech population of great reed warblers (1) there is a significant, albeit small, increase in the genetic contribution to spring as opposed to autumn migration timing, (2) genetic polymorphisms in clock-linked genes are not associated with migratory chronotype, and (3) additional candidate genes that are associated with migratory chronotype appear to participate in lipid metabolism and epigenetic processes. Future research that examines the functional aspect of these candidate genes through experimental assays, such as using transcriptomic or epigenetic techniques, will be useful for better characterizing their role in regulating migratory chronotype. For example, further insight could be gained from common garden experiments where fat reserves, intensity of lipogenesis, migratory restlessness, and methylation levels between early and late chronotype individuals are measured. Because it appears that migratory phenotypes in great reed warblers and other birds are determined by many loci of small effect, additional studies in not just other species, but multiple

populations within species are warranted (Dor et al., 2011; Peterson et al., 2013; Bazzi et al., 2015). Therefore, in the future, we encourage investigations of other populations of great reed warblers to see if similar genes are associated with the migratory phenotypes.

This study explored whether loci-specific genetic differentiation (i.e. regions under positive selection) connected to migratory timing exists in the Czech great reed warbler population, which could provide insights into how genetic factors might influence the species' adaptability to shifting environmental conditions caused by climate change. According to the IUCN Red List, the great reed warbler conservation status is of least concern, but their populations are in decline (BirdLife International, 2017). Long-distance migratory birds like great reed warblers often rely on several different habitats throughout the year (a breeding habitat, stopover habitats, and multiple wintering habitats), which increases their vulnerability because these different environments are crucial for their survival (Koleček et al., 2018; Emmenegger et al., 2021). With habitat loss and both environments changing due to climate change, it is essential that we continue to explore the genetics behind migration timing. Understanding variation in these genes could offer valuable insights into how the species might adapt to changing environmental conditions in the future.

## MATERIALS AND METHODS

### Sample collection and DNA extraction

From 2012 to 2019, a great reed warbler population was studied in two pond systems between Hodonín (48° 51′ N, 17° 07′ E) and Mutěnice (48° 54′ N, 17° 02′ E), which both lie in the southeastern region of the Czech Republic. The fieldwork was carried out with permissions of regional conservation authorities (permit nos JMK 23530/2011 and JMK 48964/2017), adhered to the Animal Care Protocol (no. 128/2010) and was in compliance with the current Czech Law on the Protection of Animals against Cruelty (license no. CZ01284). Mist nests were used to capture the warblers and blood samples were taken from the brachial vein and stored in 96% ethanol. Leg-loop mounted light-level geolocators were utilized to track the warbler's migratory movements and determine their migratory chronotype. Details on light-level data analysis, the determination of departure dates, and a detailed map of migratory routes and collection points are given in the previous studies by Koleček et al. (2016) and Pozgayová et al. (2022). Out of the 54 warblers with geolocators, 17 of them were female and 37 were male (Pozgayová et al., 2022). The blood samples were stored in a −20°C freezer prior to DNA extraction. DNA extraction was conducted using a Quick-DNA Miniprep Plus Kit (Zymo Research, Irvine, CA, USA) following the manufacturer's instructions. The final elution step was performed twice to increase DNA yield. The DNA extractions were then quantified using a Qubit 1X dsDNA HS Assay Kit (Invitrogen, Waltham, MA, USA) and stored at −20°C.

### Pool-seq library preparation and sequencing

To identify candidate genes that influence the migratory timing of great reed warblers, whole-genome sequencing was performed on both early and late migrating birds in both spring and autumn migrations. Whole-genome sequencing on many individuals is costly, so we utilized pool-seq, a cost-effective method that pools together DNA from multiple individuals within a group of interest (i.e. population) followed by whole-genome sequencing on the separate pools (Schlötterer et al., 2014; Guirao-Rico and González, 2021). Pool-seq allows the identification of single nucleotide polymorphisms (SNPs) throughout the entire genome; however, all individual genotypes are lost (Schlötterer et al., 2014). This means that when you analyze the sequencing results, you cannot differentiate one individual's sequences from another because their DNA was pooled prior to sequencing.

A total of four pools were created: early spring, late spring, early autumn, and late autumn. To make the spring pools, DNA from nine of the earliest spring migrants were pooled in equimolar ratios into one pool and DNA from nine of the latest spring migrants into a second pool (Fig. 1). When choosing which individuals to pool, the migratory chronotype, departure year, and sex ratio were all taken into consideration (Table S1). A bird's departure date for the spring migration was the date it left the last stationary non-breeding site in sub-Saharan Africa. An individual was considered to have an early chronotype if it was the earliest or one of the earliest birds to depart from the wintering site for that year. An individual was considered to have a late chronotype if it was the latest or one of the latest birds to depart from the wintering site for that year. Due to geolocators dying throughout the year, spring migratory chronotype data was only available for 40 out of the 54 birds with geolocators. Therefore, the individuals chosen for the two spring pools were selected from a list of 40 individuals (11 females and 29 males). Each spring pool had three females and six males. This sex ratio was chosen because there was a limited number of females (n=11) that had geolocator data. We only wanted to pick the earliest and latest departing birds for our pools, so having only 11 females limited the number of individuals we could choose from. Additionally, there were more males (n=29) with geolocator data, which allowed us to select more individuals with early and late chronotypes. To account for weather conditions and annual variation which could potentially affect migratory start date, birds were chosen from a variety of different years. To choose which individuals to pool, the males and females were separated. For every year where there was enough geolocator data to determine migratory chronotypes, the earliest and latest departing male and female were chosen for the early and late spring pool. If there were not six males and three females after selecting the earliest and latest male and female from each year, the second earliest or latest male or female from the year with the most individuals/geolocator data was chosen. Each spring pool had two birds whose departure year was 2013, one bird from 2015, four birds from 2016, one bird from 2017, and one bird from 2018 (Fig. 1).

During the spring migration, great reed warbler males depart from Africa before the females (Table S1; Briedis et al., 2019). The six males in the early spring pool had departure dates that ranged from the 87th to 93rd day of the year. Meanwhile, the three females in the early spring pool had departure dates that ranged from the 92nd to the 107th day of the year. A similar pattern was seen in the late spring pool. The six males in the late spring pool had departure dates that ranged from the 105th to 129th day of the year. Meanwhile, the three females in the late spring pool had departure dates that ranged from the 113th to the 133rd day of the year. Despite the differences amongst the sexes, only the earliest and latest individuals for each sex, while also considering departure year, were chosen for each pool. Therefore, it does not matter that one of the males in the late spring pool departed earlier than one of the females in the early spring pool because the sexes were separated when choosing the earliest and latest individuals.

For the two autumn pools, a similar pool-seq methodology as described above was utilized. However, because the autumn migration occurs after the breeding season, the breeding success and the number of nests each warbler had was taken into consideration. Only individuals who had only one known nest and were successful breeders (offspring had fledged) were pooled. This is because during the autumn migration, whether the individual had only one known nest and whether they successfully bred affects when they start their autumn migration (Mitchell et al., 2012; Catry et al., 2013; van Wijk et al., 2017; Chmura et al., 2020; Imlay et al., 2021). Therefore, only individuals who had only one known nest and were successful breeders were considered when forming the autumn pools. After narrowing down the individuals that could be pooled, the same pooling method that was used in the spring pools was utilized for the autumn pools (Fig. 1). A bird's departure date for the autumn migration was the last date the bird was at the last stationary period at the breeding site. An individual was considered to have an early chronotype if it was the earliest or one of the earliest birds to depart from the breeding site for that year. An individual was considered to have a late chronotype if it was the latest or one of the latest birds to depart from the breeding site for that year. The individuals chosen for the autumn pools were selected from the list of 54 individuals (17 females and 37 males) with geolocator data. Both autumn pools consisted of nine individuals; however, the early autumn pool had three females and six males whereas the late autumn pool consisted of four females and five males.

Biology Open

The sex ratio was different in the late autumn pool compared to the other three pools because of the strict criteria that the individual had to have a certain chronotype, had only one known nest, and was a successful breeder. The early autumn pool had four birds whose departure year was 2014, two birds from 2015, one bird from 2016, and two birds from 2017. The late autumn pool had one bird whose departure year was 2012, four birds from 2014, two birds from 2015, one bird from 2016, and one bird from 2017 (Fig. 1).

Unlike in the spring migration, during the autumn migration, great reed warbler males did not depart before the females (Table S1; Briedis et al., 2019). The six males in the early autumn pool had departure dates that ranged from the 184th to 202nd day of the year. Meanwhile, the three females in the early autumn pool had departure dates that ranged from the 182nd to 203rd day of the year. A similar pattern was seen in the late autumn pool. The five males in the late autumn pool had departure dates that ranged from the 203rd to the 222nd day of the year. Meanwhile, the four females in the late autumn pool had departure dates that ranged from the 214th to 224th day of the year. Once again, when choosing what individuals to pool, the females and males were separated and then chosen for the early and late autumn pool based off their chronotype. When comparing the two spring pools and two autumn pools, there were only five individuals (ZA48338, DK13980, ZA24514, ZA27389, and ZA43571) who were in both an autumn and spring pool. For most of the individuals that we chose to pool, they did not have an early or late chronotype in both the spring and in the fall. For example, bird ZA24479 was the earliest male to depart in 2016 for the spring migration; however, he was not the earliest male to depart in 2015 for the autumn migration, so he was not included in the early autumn pool. Additionally, as mentioned above, we did not have the geolocator data for all 54 individuals during the spring migration. Therefore, for 14 birds, we knew their autumn migratory chronotype, but not their spring migratory chronotype.

For each pool, a 350-bp, PCR-free library was prepared and sequenced on a NovaSeq 6000 (Illumina Inc., San Diego, CA, USA) using 2×150 paired-end chemistry (Novogene Corporation Inc., Sacramento, CA, USA). The PCR-free library preparation was specifically chosen to avoid amplification effects that may bias the inference of SNP allele frequencies.

The program *fastp* v0.20.0 (Chen et al., 2018) was used to remove sequencing adapters, bases with a quality [Q] score ≤20 from the 5′ and 3′ ends, bases from the 3′ end using a 4-bp sliding window and mean Q score ≤20, reads with >30% uncalled (N) bases, reads with a mean Q<20, reads with a complexity score <30, and reads with a final length <50 bp. The cleaned reads were mapped to the great reed warbler reference genome ASM1339868v1 (GenBank Accession Number GCA_013398685.1; Feng et al., 2020) with *BWA* v0.7.17 (Li and Durbin, 2009). Duplicated reads were removed, and only reads with a mapping quality ≥20, properly paired, and correctly oriented were retained using *SAMtools* v1.18 (Li et al., 2009).

### Pool-seq analyses

The pool-seq analyses were performed using *PoPoolation2* v1.201 (Kofler et al., 2011). First, the BAM files of aligned reads were used to generate an mpileup containing all four pools (early spring, late spring, early autumn, and late autumn) with *Samtools* and converted into the pool-seq synchronized format with *PoPoolation2* and excluding bases with a Q<20. Allele frequencies and the classical $F_{ST}$ estimator were calculated and compared pairwise between pools using the *snp-frequency-diff.pl* and *fst-sliding.pl* scrips in *PoPoolation2*, respectively. A minimum allele count of two (–min-count 2) and minimum SNP coverage of five (–min-coverage 5) were used to reduce the effect of sequencing errors, a maximum coverage of 57 (–max-coverage 57; 2.5X the mean depth of coverage per pool) was used to avoid SNPs in repetitive regions. The allele frequency and $F_{ST}$ metrics were calculated per site (–min-covered-fraction 1 –window-size 1 –step-size 1) and the pool size was set to the number of chromosomes (–pool-size 18). A Fisher's exact test of allele frequency differences per site between pools was calculated using the *fisher-test.pl* with parameters as above. From the resulting $F_{ST}$ values per site, *GenWin* v1.0 (Beissinger et al., 2015) was utilized to infer window sizes in R v4.2.1 (R Core Development Team, 2023). Many studies use rather arbitrarily defined window lengths,

potentially causing a meaningful genomic region to be artificially split into two separate windows and thus overlooked (Beissinger et al., 2015; Sly et al., 2022). To prevent this, *GenWin* was utilized to make windows based on inflection points from a fitted smoothing spline model (Beissinger et al., 2015; Sly et al., 2022).

### Quantifying the seasonal genetic contribution

To determine whether the genetic contribution of the spring migration was larger than the autumn migration, both the genome-wide, per-site $F_{ST}$ values and allele frequency differences were compared between the (1) early spring versus late spring and (2) early autumn versus late autumn pools. All metrics were calculated per site using *PoPoolation2* as described above for each pool and compared with a Z-test in R package *BDSA* v1.2.2 (Arnholt and Evan, 2023). The Z-test is the preferred test for comparing means among two groups when the sample sizes are large (>30) and population standard variation known (Pandis, 2015).

### Candidate genes associated with spring migratory chronotype

Similar to Sly et al. (2022), putative outlier windows between the early spring pool and the late spring pools were identified as those windows containing at least two significantly differentiated SNPs from the Fisher's exact test after correction for multiple comparisons using a false discovery rate (FDR) <0.01 (Benjamini and Hochberg, 1995). Using *BEDTools* v2.27.1 (Quinlan and Hall, 2010) and the reference genome ASM1339868v1 annotation in NCBI, genes within 25 kbp of each outlier window were considered significantly different between the early and late spring pools and were therefore considered as candidate genes (Sly et al., 2022). The accession number for these candidate genes were converted to UniProt IDs (Bateman et al., 2023) and put into PANTHER v.19.0 (released 2024-06-19; Thomas et al., 2022). The PANTHER *Acrocephalus arundinaceus* (great reed warbler ACRAR) reference proteome and corresponding Gene Ontology (GO) terms were used to complete a statistical overrepresentation test. All five annotation data sets in PANTHER (PANTHER Pathways, PANTHER GO-Slim Molecular Function, PANTHER GO-Slim Biological Process, PANTHER GO-Slim Cellular Component, and PANTHER Protein Class) were assessed with the Fisher's exact test and overrepresented GO terms with FDR <0.05 were retained.

### Acknowledgements

We are grateful to the Coombs high-performance computing resources available at UCF. We also thank Jaroslav Koleček for geolocator data analysis, all the people who assisted with the fieldwork, and members of the Fitak Integrative Genomics Lab at UCF for comments on earlier versions of this manuscript.

### Competing interests

The authors declare no competing or financial interests.

### Author contributions

Conceptualization: E.R.F., D.K., P.P., R.R.F.; Data curation: E.R.F., D.K., P.P., R.R.F.; Formal analysis: E.R.F., D.K., P.P., R.R.F.; Funding acquisition: E.R.F., P.P.; Investigation: E.R.F., D.K., P.P., R.R.F.; Methodology: E.R.F., D.K., P.P., R.R.F.; Writing – original draft: E.R.F.; Writing – review & editing: E.R.F., D.K., P.P., R.R.F.

### Funding

This work was funded by an award from the University of Central Florida Office of Undergraduate Research to E.R.F. P.P. was supported by the Institutional Research Plan (RVO: 68081766). Open Access funding provided by University of Central Florida. Deposited in PMC for immediate release.

### Data and resource availability

All raw sequencing data used in this study had been previously deposited in the NCBI Sequence Read Archive (https://www.ncbi.nlm.nih.gov/sra) under BioProject accession PRJNA995180. All computer code necessary to recreate the finding is available at https://github.com/rfitak/Warbler-poolseq. All relevant data and details of resources can be found within the article and its supplementary information.

### First Person

This article has an associated First Person interview with the first author of the paper.

**Peer review history**
The peer review history is available online at https://journals.biologists.com/bio/lookup/doi/10.1242/bio.062039.reviewer-comments.pdf

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
