## [Peer Review File · Biology Open]

Elucidating the genetic architecture of migratory timing in a songbird migrant, the great reed warbler, *Acrocephalus arundinaceus*

Dmitry Kishkinev, Petr Procházka, Robert R. Fitak and Emily Fackler
DOI: 10.1242/bio.062039

Editor: Lewis Halsey

Review timeline

Original submission:	28 April 2025
Editorial decision:	4 May 2025
First revision received:	1 October 2025
Accepted:	2 October 2025

Original submission

First decision letter

MS ID#: bio.062039

MS TITLE: Elucidating the genetic architecture of migratory timing in a songbird migrant, the great reed warbler, *Acrocephalus arundinaceus*

AUTHORS: Emily Fackler; Dmitry Kishkinev; Petr Procházka; Robert R. Fitak

I have now reached a decision on the above manuscript.

The reviewer reports are shown at the bottom of this email or can be accessed, together with a copy of this decision letter, by going to:

As you will see, the reviewers, particularly reviewer 1 raised a number of substantial criticisms that prevent me from accepting the paper at this stage. I'm not sure whether it is possible for you to address the biggest concerns regarding for example sample size and the lack of justification for the statistical approach; on the other hand, both reviewers see value in the study findings and indeed feel that potentially even more could be gleaned from them for example with regards possible sex differences. I invite you, therefore, to consider editing and resubmitting, and if you do your manuscript will be sent for further peer review.

At this stage, we also ask you to ensure your manuscript complies with our formatting guidelines. Provided you are able to fully address the referees' comments, we are positive about publication of your paper (we accept over 95% of revision submissions) and therefore hope you won't mind any extra work involved in reformatting your manuscript at this point.

Please ensure that you clearly highlight all changes made in the revised manuscript. Please avoid using 'Tracked changes' in Word files as these are lost in PDF conversion.

I should be grateful if you would also provide a point-by-point response detailing how you have dealt with the points raised by the reviewers in the 'Response to Reviewers' box. Please attend to all of the reviewers' comments. If you do not agree with any of their criticisms or suggestions please explain clearly why this is so.

Reviewer 1

Comments for the author

Fackler and colleagues brought interesting results on the potential genetic regulation of the migratory behavior in the great reed warbler. Overall, the paper is well written and well presented. The methods were appropriately applied with appropriate statistical support. However, given the amount of data, the author could explore some aspects of their findings. Also, as their discussion clearly states, biologically speaking, it is less probable that there are significant differences between spring vs autumn only based on F_{ST} values. The author should clearly state that in the abstract and the results. Below are some suggestions:

Quantifying the seasonal genetic contribution

The authors clearly explained the results. However, the authors could be clearer with the "very small" difference between spring and autumn, because as they briefly state, even though the stats are significant, maybe there is no biological difference. I think the authors could complement with a PCA plot that could show the separation or lack thereof of the samples. This strategy could complement the clustering of the population's polymorphisms.

Candidate genes associated with spring migratory chronotype

This point is also very well written, however, because the authors have so much data they could bring more information. Enrichment analysis is only one way to target potential functions that stand out from the expected. I encourage the authors to provide more details on the other genes, maybe through COG and KEGG terms. Or even, what is their overall function? Perhaps, the control is through expression level; do they know when and in which tissues those genes are expressed? Is there RNAseq available for that? What is the impact of the SNP/alleles in the coding genes? Is it possible to detect specific modifications in each population?

Did the authors try to select outliers in autumn early x late, to check if the 93 outliers are consistently present in both comparisons? It is not clear why only spring outliers were compared. Please provide more information/justification on that. Lastly, are those genes under selection? Could the authors test? That also could bring more information on the importance of these SNPs in the selected outliers.

The authors cited the *Dicer1* gene in the discussion, but I believe that it can be in the results too. Overall, the authors could bring more information on the 93 genes, which they only cite in the discussion.

Minor:

I don't think Figure 2 is needed. It can be added as Supplementary material.

Do the authors have a map with the migratory route and the points of collection? It would be interesting as a complement to Figure 1.

Reviewer 2

Comments for the author

General comments:

The authors of this study seek to understand the genes that influence differences in spring and autumn migration departure date among individual Great Reed Warblers from one breeding population in the Czech Republic. To do so, they re-sequence whole genomes of approx. 34 individuals with either "early" or "late" departure times (called "chronotypes"). They then use F_{ST} to compare differences between chronotypes and examine possible candidate genes underlying migratory chronotype. While the broad topic of this paper (i.e. genetic differences influencing migration timing) is exciting, the existing manuscript has several big weaknesses and points of confusion that need to be addressed. The sample sizes are fairly small for a study seeking to elucidate broad genetic differences within a breeding population.

A central weakness relates to the authors' unstated assumption that there are genetic differences between migratory chronotypes (and subsequent analytical approach). No genetic data are provided to substantiate this claim. Yet, the authors analyze differences in chronotypes using F_{st} , which is an index that measures differentiation due to genetic structure. F_{st} is a population-level statistic, so its use only makes biological and statistical sense if the studied migratory chronotypes exhibit structure. In the introduction, the authors state that populations of Great Reed Warblers differ in migration timing (which implies potential pop structure), but in this study, only breeders from the Czech Republic population were sampled. The authors additionally state that individuals within populations may have flexible departure dates (Lines 75-78). Thus, it is very odd and not clear why the authors use pairwise F_{st} to examine differences among the selected individuals for three main reasons: 1) we would not predict genetic structure among individuals from the same breeding population; 2) F_{st} therefore is not the appropriate statistic for the comparisons in this study; and 3) because differences in migration timing may be seasonally or annually plastic (see below).

In the introduction, the authors provide citations and somewhat superficial discussion of 'migration genes' underlying departure timing differences. While candidate genes underlying migration have certainly been documented, migration timing is a complex phenotype that is unlikely to be explained entirely by one or a handful of genes. The framing of this paper could be strengthened by including additional papers that reference the complex nature of migration. Additionally, the introduction would be made more robust by incorporating the alternative hypothesis that migration departure may be shaped simultaneously by genetic cues *and* seasonal and/or annual environmental differences.

More broadly, the introduction is missing citations to a few relevant Great Reed Warbler publications (e.g., Sjoberg et al. 2018 *J. Avian Biol.*, Liechti et al. 2018 *Movement Ecology*).

The authors should address known sex differences in migration departure dates - these are not mentioned but are evident in Table S1 (i.e., males consistently depart earlier than females in the spring. For example, "late" spring male ZA27389 leaves on day 105, which is 2 days before "early" spring females Z784791 and ZA18222). Despite this, the authors combine males and females into the same chronotype (in uneven sampling numbers). There is an additional interesting pattern which the authors do not discuss, which is that the aforementioned trend flip-flops between spring and autumn: In autumn, females appear to depart earlier than males. Why? Mentioning sex differences in departure dates, seasonal shifts in departure patterns, and providing justification for why all individuals were analyzed in the same chronotype groups despite sex differences in departure dates is necessary.

Related to this, many methods details in this manuscript need to be clarified - see below for specific comments. In particular, it is not clear how many individuals the authors sequenced (stated 9 individuals/pool x 4 chronotypes does not sum to the 54 birds described to be tracked with geolocators, which are said to be represented in Fig. 1). Sex ratio choices, selection of years, and why the authors chose to use different sets of birds to compare spring and autumn migration timing, rather than comparing differences in spring and autumn dates for the *same* individuals need to be justified (only 5 individuals were included in both pools - why?)

How were "early" and "late" departures defined? The authors use these terms but never provide a strict definition about what these terms mean, particularly in relationship to one another.

Finally, given assumptions made about genetic structure and the current F_{st} approach used, the authors should present, at minimum, a PCA plot showing clustering of sampled individuals to verify that chronotypes do (or do not) cluster together.

Specific comments:

Lines 49-57: In some systems migratory timing may be plastic and determined by availability of seasonal resources. The authors should provide support for this alternative hypothesis.

Line 66: Fix singular/plural disagreement in this line

Lines 82-85: This point should be expanded and thought through more. There is likely to be little reproductive advantage to autumn migration departure date, though carryover effects may impact reproduction in the following year. Individuals may also be exposed to fluctuating or varying resource along the migratory route, favoring earlier departure, for example, to capitalize on resources pulses.

Lines 85-89: This is not unique to Great Reed Warblers - it is well established that spring migration occurs much faster than autumn migration.

Lines 89-92: Clarify what you mean in this sentence.

Lines 97-109: This paragraph superficially touches on the idea of "migration genes" but does not adequately address that migration involves a complex suite of interacting genes and environmental cues. It also does not fully cite relevant papers from other systems.

Lines 110: This assumes that there is a 'migration gene' for migratory chronotype (which this reviewer does not buy). It would be more robust to phrase this as something more in the vein of: "we do not yet know if migratory chronotype has a genetic basis".

Line 111: What the authors mean by "genetic contribution" is not clear. (Proportion of genes? A certain set of genes? Genes under selection? Fixed differences between/among populations?)

Lines 127-131: Sex differences in timing are not discussed, nor is the 'flip-flop' pattern in departure date of sexes between spring and autumn.

Lines 136-137: It seems a bit odd that depth of coverage decreased so substantially after filtering. What was mean depth of coverage per genome?

Lines 223-224: Was fat noted on captured/handled individuals?

Lines 227-229: This claim does not seem substantiated by these data - tone this down.

Lines 229-235: The authors do not address the alternative hypothesis that fat deposition is dependent upon seasonal resources and individual foraging success; thus, annual environment may determine fat loading, and thus predict migration timing.

Lines 232-233: Do the authors have multiple years of tracking data? This would provide some insight into how flexible interannual migration departure dates are for individuals.

Lines 306-309: This phrasing is somewhat different than the stated goal throughout this paper. I would argue that the authors should provide this data to substantiate their analyses - see many comments about this above. Additionally, these data are not presented, so it is not possible to evaluate genetic structure (Fig 1 of F_{st} does not elucidate structure).

Lines 307-317: Climate change link in conclusions feels a bit tacked on, particularly since it isn't integrated into the fabric of the study.

Lines 320-337: How many total warblers were sampled for this study? And how many of each chronotype? How were chronotypes determined?

Lines 346-347: Clarify what you mean by "all genotypes lost".

Line 349: "Migrator" is a somewhat unusual way to say "migrant"; use the latter.

Lines 348-357: This section is missing necessary details. Please clearly describe the number of individuals per each spring and autumn pools. The authors mention balancing sex ratios but subsequently say that spring pools had 3 females and 6 males, which is not a balanced sex ratio. Presumably, migration departure dates were determined using geolocator data - this should be confirmed. How were "stationary non-breeding sites" determined? Describe methods/thresholds. By sequencing birds across many different years (and without presenting data about inter-annual

consistency in migration timing, particularly among individuals known to differ in their annual departure times - as stated in the introduction), the authors have potentially introduced year as a covariate - this should be justified. Additionally, please explain why these particular years were chosen (skipping 2014 and 2019; and sampling from different numbers of individuals per year). Why didn't the authors just choose the earliest-departing and latest-departing individuals? This point becomes more confusing when viewing Figure 1, which shows a clear gradient in migration departure timing.

Line 402: State whether weighted F_{st} (this is what should be used; though see above about how this approach seems inappropriate for the current questions).

Line 410: Still unclear what "genetic contribution" is.

Figures:

Figure 1: In caption, change "their" to "its". Caption states that 54 geolocator-tracked birds are represented but only 34 vertical lines appear. Clarify that x-axis is Julian date. How were "early" and "late" departures defined? This definition is critical for this paper, and essential to explain, particularly given what appears to be a fairly predictable gradient in migration departure timing (e.g., the vertical yellow line right within about a day of the vertical green line for fall migration; those two birds departed at essentially the same time, but one is called "early" and one is called "late").

Figure 2: This is more of a methods figure (i.e., not essential for results) that should be moved to the supplement.

Figure 3: Clarify which scaffold/set of scaffolds on the x-axis. Clarify that this is a pairwise comparison. Why isn't fall comparison shown? This would be best as a multi-panel.

Reviewer's Responses to Questions

Experimental quality

Does each figure have the proper controls?

If 'No', please indicate reasons in Comments for Author box below.

Reviewer #1:

- Yes

Reviewer #2:

- Yes

Were the data analyzed using appropriate statistical tests?

If 'No', please indicate reasons in Comments for Author box below.

Reviewer #1:

- Yes

Reviewer #2:

- No

Reproducibility

Were experiments performed using adequate number of biological replicates?
If 'No', please indicate reasons in Comments for Author box below.

Reviewer #1:

- Yes

Reviewer #2:

- No
-
-

Does the methods section provide sufficient detail to permit reproducibility?
If 'No', please indicate reasons in Comments for Author box below.

Reviewer #1:

- Yes

Reviewer #2:

- No
-
-

Completeness

Are the manuscript's conclusions supported by the data?
If 'No', please indicate reasons in Comments for Author box below.

Reviewer #1:

- No

Reviewer #2:

- Yes
-
-

Scholarship

Do the authors cite and discuss the merits of data that would argue for and against their conclusion?
If 'No', please indicate reasons in Comments for Author box below.

Reviewer #1:

- Yes

Reviewer #2:

- No

Does the manuscript title & abstract accurately reflect the contents of the manuscript, without hyperbole?

If 'No', please indicate reasons in Comments for Author box below.

Reviewer #1:

- No

Reviewer #2:

- No

First revision

Author response to reviewers' comments

Reviewer 1:

Fackler and colleagues brought interesting results on the potential genetic regulation of the migratory behavior in the great reed warbler. Overall, the paper is well written and well presented. The methods were appropriately applied with appropriate statistical support. However, given the amount of data, the author could explore some aspects of their findings. Also, as their discussion clearly states, biologically speaking, it is less probable that there are significant differences between spring vs autumn only based on FST values. The author should clearly state that in the abstract and the results. Below are some suggestions:

Quantifying the seasonal genetic contribution

The authors clearly explained the results. However, the authors could be clearer with the "very small" difference between spring and autumn, because as they briefly state, even though the stats are significant, maybe there is no biological difference.

Lines 31-32 & 165-169: We appreciate this suggestion and made this clearer in both the abstract and results.

I think the authors could complement with a PCA plot that could show the separation or lack thereof of the samples. This strategy could complement the clustering of the population's polymorphisms.

We appreciate the comment about adding a PCA plot to the paper. However, because we used the pool-seq method, individual genotype information is not available (see lines 396-405 which describe this method). Because there are only four pools in our study, there would only be four points on the PCA plot and we thus would not provide any additional insight or information to the reader.

Candidate genes associated with spring migratory chronotype

This point is also very well written, however, because the authors have so much data they could bring more information. Enrichment analysis is only one way to target potential functions that

stand out from the expected. I encourage the authors to provide more details on the other genes, maybe through COG and KEGG terms. Or even, what is their overall function?

We appreciate the suggestion by the reviewer. We already performed and included enrichment tests for all five annotation data sets in PANTHER (PANTHER Pathway, PANTHER GO-Slim Molecular Function, PANTHER GO-Slim Biological Process, PANTHER GO-Slim Cellular Component, and PANTHER Protein Class) on the 93 candidate genes (lines 177-186). Notably, there are three reasons why the suggested COG/KEGG tests are not necessary. First, the PANTHER database is perhaps one of the most comprehensive databases of protein/gene function, ontology, pathways and statistical analysis tools, and the PANTHER Pathways module meets Systems Biology Graphical Notation Process Description standards (see doi: [10.1038/nprot.2013.092](https://doi.org/10.1038/nprot.2013.092)). Second, PANTHER Pathways assignments are analogous to KEGG/COG pathways, thus the requested tests would be largely redundant. Third, because great reed warblers are a non-model species, we are fortunate to have their proteome already curated in PANTHER but not in COG/KEGG. Although we could adapt the warbler proteome pathways assignments to KEGG via homology, this would arguably be less robust than their current, well-curated assignments in PANTHER.

As a result, we do not believe that COG and KEGG terms would provide any additional information that is not already listed in our paper or supplementary information from the overrepresentation test; however, we do agree that more information regarding the candidate genes could be put into the paper. Please see the lines 186-198 for additional gene information that has been added. For additional functional information on the 93 candidate genes, please see supplementary information table 2 which has all the PANTHER gene family IDs assigned in addition to other functional information.

Perhaps, the control is through expression level; do they know when and in which tissues those genes are expressed? Is there RNAseq available for that? What is the impact of the SNP/alleles in the coding genes? Is it possible to detect specific modifications in each population?

We agree with the reviewer, that indeed, gene expression and/or other mechanism of phenotypic plasticity likely play a role. We have added this to future directions in the conclusion (lines 350-353). The 93-candidate genes could absolutely affect gene expression. However, in this study we aimed to identify evidence of selection. An analysis of gene expression is beyond the scope of this study since it would likely require more invasive or even lethal sampling and knowledge of the specific tissues involved, such as neurological tissue. Furthermore, no comprehensive gene expression atlas across tissues in the great reed warbler or close relatives exists.

Did the authors try to select outliers in autumn early x late, to check if the 93 outliers are consistently present in both comparisons? It is not clear why only spring outliers were compared. Please provide more information/justification on that. Lastly, are those genes under selection? Could the authors test? That also could bring more information on the importance of these SNPs in the selected outliers.

We restricted our test for outliers only to the early vs late spring comparison. We included the autumn comparison to test only the hypothesis that there was a larger effect in spring vs autumn. Our second hypothesis regarding genes potentially under selection was restricted to spring comparison only, and performing outlier tests for the autumn will only decrease our power and provide no relevant information (whether or not outliers are detected) regarding the hypothesis presented. Additionally, arriving to the breeding site early is shown to lead to higher fitness while in comparison, there is weak evidence for the adaptive value of autumn departure and the arrival to wintering grounds. For the second point, we confirm that indeed, we already completed tests for selection, and these 93 outliers fit our criteria for positive selection as described in the manuscript.

The authors cited the Dicer1 gene in the discussion, but I believe that it can be in the results too. Overall, the authors could bring more information on the 93 genes, which they only cite in the discussion.

Lines 186-198, 259-267, & 327-343: We agree with the reviewer. More information about the 93 candidate genes were added to both the results and discussion.

Minor:

I don't think Figure 2 is needed. It can be added as Supplementary material.

Figure 2 was removed from the paper and added to the supplementary material.

Do the authors have a map with the migratory route and the points of collection? It would be interesting as a complement to Figure 1.

We appreciate this suggestion. A map of this exact route and collection points is already available in co-author Procházka's article (Koleček et al. 2016). Although we already cite this work, we have added a statement (lines 385-388) to specifically indicate that a map is available at this reference.

Reviewer 2:

General comments:

The authors of this study seek to understand the genes that influence differences in spring and autumn migration departure date among individual Great Reed Warblers from one breeding population in the Czech Republic. To do so, they re-sequence whole genomes of approx. 34 individuals with either "early" or "late" departure times (called "chronotypes"). They then use *Fst* to compare differences between chronotypes and examine possible candidate genes underlying migratory chronotype. While the broad topic of this paper (i.e. genetic differences influencing migration timing) is exciting, the existing manuscript has several big weaknesses and points of confusion that need to be addressed. The sample sizes are fairly small for a study seeking to elucidate broad genetic differences within a breeding population.

We appreciate the feedback from the reviewer, and agree that the sample size is not large. This small size was necessary because we carefully selected individuals from a larger group that met very specific criteria (see updated sampling description lines 409-410, 419-422, 444-453). These criteria included taking chronotype, departure year, sex, and nesting success into consideration to account for weather conditions and annual variation which could potentially affect migratory start date. Although a smaller sample size, this set of 9 birds per pool is highly curated to be matched and comparable. If we had selected a larger pool size, variability due to the factors above rather than just chronotype would obscure the results.

We have also added additional discussion (lines 227-230) to communicate to the reader that our results are likely not identifying all loci under selection, as there may be more loci of weaker effect that would be detected with larger sample sizes.

A central weakness relates to the authors' unstated assumption that there are genetic differences between migratory chronotypes (and subsequent analytical approach). No genetic data are provided to substantiate this claim. Yet, the authors analyze differences in chronotypes using *Fst*, which is an index that measures differentiation due to genetic structure. *Fst* is a population-level statistic, so its use only makes biological and statistical sense if the studied migratory chronotypes exhibit structure. In the introduction, the authors state that populations of Great

Reed Warblers differ in migration timing (which implies potential pop structure), but in this study, only breeders from the Czech Republic population were sampled. The authors additionally state that individuals within populations may have flexible departure dates (Lines 75-78). Thus, it is very odd and not clear why the authors use pairwise *Fst* to examine differences among the selected individuals for three main reasons: 1) we would not predict genetic structure among individuals from the same breeding population; 2) *Fst* therefore is not the appropriate statistic for the comparisons in this study; and 3) because differences in migration timing may be seasonally or annually plastic (see below).

We apologize for the lengthy response here but must clarify some questions presented by the Reviewer. We highly encourage the Editor to carefully consider our response regarding the fundamentals of *Fst* metrics.

First, the Reviewer suggests that we have some unstated assumptions regarding genetic differences among chronotypes. We respectfully disagree, since the null hypothesis of the Fst-outlier approach is that indeed, there are no genetic differences (i.e. $F_{st} = 0$ or no difference in allele frequencies) among the two groups compared. We think the confusion may arise from the Reviewer being concerned about genome-wide differences in Fst, which would be caused by demographic processes such as population structure, whereas an Fst-outlier approach is used to detect not genome-wide differences, but rather single-locus deviations resulting from selection. As the Reviewer is aware, selection (in this case directional or disruptive) is acting upon single loci and not the entire genome.

Second, we would like to clarify the use of the Fst statistic. But first, we do state that in addition to Fst, we applied a straightforward statistical test of allele frequency differentiation to further support our results, thus they do not rely solely on Fst. At its core, Fst measures the reduction in heterozygosity due to population subdivision (originally defined by Wright 1943). There is no assumption when using Fst that any two demes being compared are required to be structured *a priori*. If there is indeed no substructure, then subsets within a deme (e.g., “subpopulations”) will not differ in their allele frequencies and $F_{st}=0$. This is a critical assumption for well-crafted association studies that link genetic variation to a phenotype of interest (e.g., GWAS). If the population were structured, then this would bias any associations that are found. In our study, it was carefully designed to interrogate a single population to avoid confounding associations due to population structure. This design ensures that any loci that depart from the genome-wide background are undergoing a non-neutral process, not the result of cryptic population structure, and is the prevailing design behind thousands upon thousands of GWAS and Fst-outlier studies published. Please see the statement below from the extensive Fst review by Holsinger and Weir (2009): *“If natural selection favours one allele over others at a particular locus in some populations, the FST at that locus will be larger than at loci in which among-population differences are purely a result of genetic drift. Genome scans that compare single-locus estimates of Fst with the genome-wide background might therefore identify regions of the genome that have been subjected to diversifying selection.”*

Third, we would like to further clarify that Fst is not a population-level statistic. It is a parameter of hierarchical population structure defined by Wright (1943) with many different estimators existing in largely 2 classes (e.g., Nei 1973 vs Weir and Hill 2002). These two classes of estimators make different assumptions about how the total population is defined. The first class of estimators (e.g., Nei 1973 or Gst), which we used (see response detailed to a different comment below) assumes the total population is the sum of the two subpopulations. Alternatively, population-specific Fst measures, such as Weir and Hill 2002, assume that the total population is the most recent ancestor population of the two demes. The former is more appropriate for our pool-seq design, and the latter more appropriate for detecting demographic structure due to drift. We encourage the Reviewer to see such summaries (Bhatia et al. 2013, Holsinger 2009, Holsinger and Weir 2009).

Lastly, to summarize our response to the 3 points raised by this Reviewer: 1) by intended design, our hypothesis assumes a single breeding population, thus we do not expect genetic structure on average across the genome. But when single loci differ from this null distribution it can be the result of selection, 2) Fst is the appropriate, and ideal statistic for identifying locus-specific departures from the null distribution, and we have concurrently also reported significant differences in allele frequencies to support our results, and 3) as we further detail below, we agree that chronotypes are very likely the result of multiple factors, such as environment and genetics. We do not attempt to explain any of the environmental variation in chronotype, but only the heritable genetic portion.

In the introduction, the authors provide citations and somewhat superficial discussion of 'migration genes' underlying departure timing differences. While candidate genes underlying migration have certainly been documented, migration timing is a complex phenotype that is unlikely to be explained entirely by one or a handful of genes. The framing of this paper could be strengthened by including additional papers that reference the complex nature of migration. Additionally, the introduction would be made more robust by incorporating the alternative hypothesis that migration departure may be shaped simultaneously by genetic cues *and* seasonal and/or annual environmental differences.

Lines 58-69: We have added more details and associated references in the introduction on how environmental factors and resource availability can influence migratory timing. We do clarify for the reviewer that we in no way state or imply that genes are the only factor influencing timing. In this study, we are only interested in elucidating the genetic component, which, of course, does not explain a chronotype in its entirety. This is currently described on lines 53-57 and 69-74.

More broadly, the introduction is missing citations to a few relevant Great Reed Warbler publications (e.g., Sjoberg et al. 2018 J. Avian Biol., Liechti et al. 2018 Movement Ecology).

Lines 78-80: These publications have been added to the introduction and references.

The authors should address known sex differences in migration departure dates - these are not mentioned but are evident in Table S1 (i.e., males consistently depart earlier than females in the spring. For example, "late" spring male ZA27389 leaves on day 105, which is 2 days before "early" spring females Z784791 and ZA18222). Despite this, the authors combine males and females into the same chronotype (in uneven sampling numbers). There is an additional interesting pattern which the authors do not discuss, which is that the aforementioned trend flip-flops between spring and autumn: In autumn, females appear to depart earlier than males. Why? Mentioning sex differences in departure dates, seasonal shifts in departure patterns, and providing justification for why all individuals were analyzed in the same chronotype groups despite sex differences in departure dates is necessary.

Lines 424-429, 432-442, 459-463, & 467-476: We have clarified in the methods that there are sex differences in departure dates, seasonal shifts in departure patterns, and provide further justification for our strict inclusion criteria that accounts for sex differences in departure dates. Additionally, the males and females were chosen separately due to between-sex differences in phenology of migration (see lines 424-429).

Related to this, many methods details in this manuscript need to be clarified - see below for specific comments. In particular, it is not clear how many individuals the authors sequenced (stated 9 individuals/pool x 4 chronotypes does not sum to the 54 birds described to be tracked with geolocators, which are said to be represented in Fig. 1). Sex ratio choices, selection of years, and why the authors chose to use different sets of birds to compare spring and autumn migration timing, rather than comparing differences in spring and autumn dates for the *same* individuals need to be justified (only 5 individuals were included in both pools - why?)

Lines 410-422, 424-429, 438-440, 455-464, & 474-484: We remind the reviewer that the study started with geolocator departure data for 54 birds, and subsets of these 54 were selected for each pool. Please see our response to this reviewer's previous comment where we elaborate in more detail on the importance of our strict selection criteria.

We have added clarification in the methods as to why the sex ratios were chosen for each pool, why we chose to incorporate multiple years of migratory departure data, why different birds were used in the spring and autumn pools, and why five individuals were in an autumn and spring pool.

How were "early" and "late" departures defined? The authors use these terms but never provide a strict definition about what these terms mean, particularly in relationship to one another.

Lines 412-415 & 453-457: We have now extensively clarified in the methods as to how the "early" and "late" departures were defined (also see our previous responses to this reviewer).

Finally, given assumptions made about genetic structure and the current Fst approach used, the authors should present, at minimum, a PCA plot showing clustering of sampled individuals to verify that chronotypes do (or do not) cluster together.

We appreciate the comment about adding a PCA plot to the paper. However, we remind the reviewer that the pool-seq method does not provide individual genotype information (see lines 396-405 which describe this method), but rather the allele frequency information for the pool. Because there are only four pools in our study, there would only be four points on the PCA plot

and thus would not provide any additional insight or information to the reader.

Specific comments:

Lines 49-57: In some systems migratory timing may be plastic and determined by availability of seasonal resources. The authors should provide support for this alternative hypothesis.

Lines 58-69: We have added more details on how environmental factors and resource availability can influence migratory timing in the introduction.

Line 66: Fix singular/plural disagreement in this line

Lines 75: The singular/plural disagreement was corrected.

Lines 82-85: This point should be expanded and thought through more. There is likely to be little reproductive advantage to autumn migration departure date, though carryover effects may impact reproduction in the following year. Individuals may also be exposed to fluctuating or varying resource along the migratory route, favoring earlier departure, for example, to capitalize on resources pulses.

Lines 58-69 & 96-98: We have elaborated on how external factors and carryover effect could influence autumn migratory chronotype.

Lines 85-89: This is not unique to Great Reed Warblers - it is well established that spring migration occurs much faster than autumn migration.

Lines 98-101: We have clarified in the introduction that this is not unique to great reed warblers.

Lines 89-92: Clarify what you mean in this sentence.

Lines 101-105: We have clarified this sentence with additional detail and split into two sentences.

Lines 97-109: This paragraph superficially touches on the idea of "migration genes" but does not adequately address that migration involves a complex suite of interacting genes and environmental cues. It also does not fully cite relevant papers from other systems.

We appreciate this comment and agree that migration is indeed a complex trait. In the beginning of this paragraph, we now address that migratory behavior is complex and likely influenced by many genes of variable effect (lines 110-111). We then narrow this to several of the candidate genes that are regularly reported to be involved. We disagree, however, with the need for relevant papers from other systems as this exceeds the narrow scope of the system investigated in this study (great reed warblers), and because this field is so large and extensive in other systems (e.g., other birds, fish, sea turtles, whales, etc...) an exhaustive description and list of numerous citations would be unnecessarily tedious for readers.

Lines 110: This assumes that there is a 'migration gene' for migratory chronotype (which this reviewer does not buy). It would be more robust to phrase this as something more in the vein of: "we do not yet know if migratory chronotype has a genetic basis".

Line 124-127: This sentence was rephrased, because like the reviewer, we also do not think there is a single migratory gene.

Line 111: What the authors mean by "genetic contribution" is not clear. (Proportion of genes? A certain set of genes? Genes under selection? Fixed differences between/among populations?)

We have carefully selected this term "genetic contribution" because it specifically can encompass each of the above possibilities. Listing each of these items is unnecessarily garrulous and we

clearly articulate in the methods and results the various genetic parameters that are measured.

Lines 127-131: Sex differences in timing are not discussed, nor is the 'flip-flop' pattern in departure date of sexes between spring and autumn.

Lines 432-442 & 467-476: We provided additional, extensive clarification in the methods section regarding the individual, sex, and chronotype differences. Particularly regarding our careful inclusion criteria from the set of 54 individual birds to construct the 4 pools.

Lines 136-137: It seems a bit odd that depth of coverage decreased so substantially after filtering. What was mean depth of coverage per genome?

We note that this proportion (~30% attrition) is quite representative when considering the numbers reflect the initial, raw data and that after the careful quality control criteria enabled to minimize false positive SNPs, such as trimming, mapping, retaining only properly paired and oriented mapped reads, etc. In other words, we could have kept more data by being less strict, but likely including many more false positive SNPs. Furthermore, the reference genome assembly is not a chromosome-level assembly. The amount of sequencing data, including raw, after filtering, and after mapping, can be found in Figure S1.

Lines 223-224: Was fat noted on captured/handled individuals?

Fat scores were noted; however, because the birds were tagged and tags retrieved during the breeding season, fat levels at deployment and retrieval were generally low. For this reason, fat scores were not informative for the scope of this study.

Lines 227-229: This claim does not seem substantiated by these data - tone this down.

We appreciate this comment from the Reviewer. We clarify that our data do show that the genes we identified are enriched for these functions and we also deliberately use the phrase "our data suggest" to articulate to the reader that this is speculation consistent with our findings, but not directly measured. This type of cautious speculation is standard in the discussion as it helps place the results in context and indicates future research directions.

Lines 229-235: The authors do not address the alternative hypothesis that fat deposition is dependent upon seasonal resources and individual foraging success; thus, annual environment may determine fat loading, and thus predict migration timing.

We appreciate this suggestion. In this part of the discussion, we articulate that this is speculation, and we indicate that this assumes "equal dietary intake". As stated above to this reviewer (currently described on lines 54-74), we do not state that genetics explains all the variation in chronotype. There is a clearly a large amount of plasticity associated (as suggested by the reviewer). Our study, rather, identifies the genetic component, even if small, relevant to the more variable environmental/individual component. Since there are an infinite number of alternative hypotheses, including this Reviewer's, that could explain the non-genetic component, and since the Reviewer's hypothesis was not in our study's design, we decide to not discuss it in this section.

Lines 232-233: Do the authors have multiple years of tracking data? This would provide some insight into how flexible interannual migration departure dates are for individuals.

We have multiple years of observational data for spring arrivals; however, this is beyond the scope of the manuscript.

Lines 306-309: This phrasing is somewhat different than the stated goal throughout this paper. I would argue that the authors should provide this data to substantiate their analyses - see many comments about this above. Additionally, these data are not presented, so it is not possible to evaluate genetic structure (Fig 1 of F_{st} does not elucidate structure).

We invite the reviewer to see our previous response regarding the term differentiation and use of F_{st} . In brief, we understand that "genome-wide" differentiation is not the goal of this study, but

specific loci in the genome with excessive divergence (i.e. F_{st} outliers or genes under selection). Therefore, we have clarified in this sentence our locus-specific intent.
 “This study explored whether loci-specific genetic differentiation (i.e. regions under positive selection) connected to migratory timing exists in the Czech great reed warbler population.”

Lines 307-317: Climate change link in conclusions feels a bit tacked on, particularly since it isn't integrated into the fabric of the study.

We include this as part the overall big picture, broader impacts of the study and to further engage in a wide audience. Because environmental cues - which as this reviewer pointed out are important for migratory timing - are susceptible to changing climate we will keep this at the end of our concluding remarks.

Lines 320-337: How many total warblers were sampled for this study? And how many of each chronotype? How were chronotypes determined?

We have clarified this in the methods, and in our previous responses. A total of 54 birds were sampled (line 388 states this number), chronotypes were determined according to strict inclusion criteria (see lines 146-149, 412-415, & 444-457), and subsets of individuals within a chronotype were combined into 4 different pools (see lines 424-431 & 457-466).

Lines 346-347: Clarify what you mean by "all genotypes lost".

Lines 403-405: We have clarified this in the methods. Notably, we must remind the reviewer here that the pool-seq technique, which mixes DNA from different individuals in the same tube, means that we cannot infer each individual's genotype (i.e., it is 'lost'), however, we can accurately infer allele frequencies.

Line 349: "Migrator" is a somewhat unusual way to say "migrant"; use the latter.

Lines 407-409: The term "migrator" was changed to "migrant".

Lines 348-357: This section is missing necessary details. Please clearly describe the number of individuals per each spring and autumn pools. The authors mention balancing sex ratios but subsequently say that spring pools had 3 females and 6 males, which is not a balanced sex ratio. Presumably, migration departure dates were determined using geolocator data - this should be confirmed. How were "stationary non-breeding sites" determined? Describe methods/thresholds. By sequencing birds across many different years (and without presenting data about inter-annual consistency in migration timing, particularly among individuals known to differ in their annual departure times - as stated in the introduction), the authors have potentially introduced year as a covariate - this should be justified. Additionally, please explain why these particular years were chosen (skipping 2014 and 2019; and sampling from different numbers of individuals per year). Why didn't the authors just choose the earliest-departing and latest-departing individuals? This point becomes more confusing when viewing Figure 1, which shows a clear gradient in migration departure timing.

As we addressed above, we have extensively clarified in the methods the number of individuals in each pool (lines 416-418 & 457-460), why the sex ratios were chosen for each pool (lines 418-422 & 457-463), why particular years were chosen (lines 422-424), and why certain individuals were chosen over others (lines 409-410, 425-429, & 450-451). To briefly explain why 2014 and 2019 were not in the study, the project lasted from 2012-2018 and 2014 was skipped for the spring migration pools due to the lack of data collected. The Figure 1 caption was updated to clarify the gradient in migration departure date (lines 841-848). As mentioned on lines 385-388, please see Kolecek et al. (2016) and Pozgayová et al. (2022) for a detailed explanation on light-level data analysis and how stationary periods were determined. Briefly, we used the criteria mentioned in the papers that assign a stopover location outside breeding site and breeding season to being stationary if a bird stays at it over certain number of days without evidence of long-distance movement.

Line 402: State whether weighted F_{st} (this is what should be used; though see above about how this approach seems inappropriate for the current questions).

We used the standard F_{st} implemented in Popoolation2 and not the weighted “Karlsson” method. We have now clarified this on lines 502-504. According to Karlsson et al. 2007 (supplementary methods), when the sample sizes for the two populations (here ‘pools’) are equal, then this reduces to the unweighted equivalent of Weir and Hill 2002. Nevertheless, in this scenario, the standard F_{st} definition in Popoolation2 is generally that of Nei 1973 (originally called G_{st}). This F_{st} formulation assumes that the “total population” is the sum of the two subpopulations. In our warbler study, this is the ideal assumption. Alternatively, population specific F_{st} measures, such as by Weir and Hill (2002) - and by inclusion Karlsson et al. 2007 - assume that the “total population” is the most recent common ancestor population of the two subpopulations. This assumption is less appropriate for our study. We encourage the Reviewer to revisit summary work by Bhatia et al 2013 (doi: 10.1101/gr.154831.113) and Holsinger 2009 (https://opencommons.uconn.edu/eeb_articles/22). Please see additional comments above for further arguments clarifying this reviewer’s misunderstanding of F_{st} .

Line 410: Still unclear what “genetic contribution” is.

Please see our comment above to this same reviewer regarding our use of the term.

Figures:

Figure 1: In caption, change “their” to “its”. Caption states that 54 geolocator-tracked birds are represented but only 34 vertical lines appear. Clarify that x-axis is Julian date. How were “early” and “late” departures defined? This definition is critical for this paper, and essential to explain, particularly given what appears to be a fairly predictable gradient in migration departure timing (e.g., the vertical yellow line right within about a day of the vertical green line for fall migration; those two birds departed at essentially the same time, but one is called “early” and one is called “late”).

Figure 1 was updated to address the above comments (lines 841-848). See the methods section for why there are not 54 vertical lines in the graph (only individuals chosen for the pools are represented as vertical lines on the graph), how early and late departures were defined, and why two birds departed at the same time, but one has an early chronotype and one has a late chronotype (they were different sexes).

Figure 2: This is more of a methods figure (i.e., not essential for results) that should be moved to the supplement.

Figure 2 was removed from the paper and added to the supplementary material.

Figure 3: Clarify which scaffold/set of scaffolds on the x-axis. Clarify that this is a pairwise comparison. Why isn't fall comparison shown? This would be best as a multi-panel.

Line 849-855 (Fig 2 caption): We have clarified in the caption that the scaffolds are arranged in numerical order from VZST01000001.1 to VZST01029468.1. Due to the large number of scaffolds in the reference genome (29,468), they are not shown individually. As stated previously to this reviewer, our study’s hypothesis only examines selection in the spring comparison, not the fall, so we are only showing the relevant pairwise comparison and not any additional panels.

Second decision letter

MS ID#: bio.062039R1

MS TITLE: Elucidating the genetic architecture of migratory timing in a songbird migrant, the great reed warbler, *Acrocephalus arundinaceus*

AUTHORS: Emily Fackler; Dmitry Kishkinev; Petr Procházka; Robert R. Fitak

I've worked through your extensive rebuttal letter this morning. I appreciate the depth of your responses and the associated edits you've made to your manuscript. Most importantly, I am satisfied with your response to Reviewer 1 regarding issues around Fst metrics. In turn I am happy to tell you that your manuscript has been accepted for publication in Biology Open, pending our standard publication integrity checks. It was accepted on 2nd October 2025.